# Effects of Molecular Weight and Degree of Esterification of Soluble Soybean Polysaccharide on the Stability of Casein under Acidic Conditions

**DOI:** 10.3390/foods10030686

**Published:** 2021-03-23

**Authors:** Hao Tian, Qizhu Zhao, Zhiyong He, Zhaojun Wang, Fang Qin, Maomao Zeng, Jie Chen

**Affiliations:** 1State Key Laboratory of Food Science and Technology, Jiangnan University, Wuxi 214122, China; 6180111134@stu.jiangnan.edu.cn (H.T.); 6180112122@stu.jiangnan.edu.cn (Q.Z.); zyhe@jiangnan.edu.cn (Z.H.); wangzhaojun@jiangnan.edu.cn (Z.W.); qfflast@sina.com (F.Q.); 2International Joint Laboratory on Food Safety, Jiangnan University, Wuxi 214122, China; 3Food Chemistry and Food Development, Department of Life Technologies, University of Turku, FI-20014 Turku, Finland

**Keywords:** SSPS, casein, complex content, molecular weight, degree of esterification

## Abstract

The effects of molecular weight (MW) and degree of esterification (DE) of soluble soybean polysaccharide (SSPS) on the stability of casein under acidic conditions were investigated. The ability of SSPS to stabilize casein was characterized by the content of SSPS–casein complex, the LUMiSizer instability index, average particle size, zeta potential, and storage experiments. The long-term storage stability of the mixtures was related to their ability to combine casein and the stability of the complexes. At the same DE, SSPSs with medium MW formed more complexes with casein than SSPSs with high or low MW; and at the same MW, SSPSs with medium or low DE formed more complexes than SPSSs with high DE. In addition, SSPSs with higher MW had a better stabilizing behavior due to the large steric repulsion between complexes. SSPSs with high MW and low DE showed the best ability to stabilize casein under acid conditions.

## 1. Introduction

Acidified milk drinks are a series of beverages produced by dilution, acidification, and homogenization, such as yogurt drinks, milk drinks with fruit, buttermilk, and whey drinks [1,2,3,4,5]. They are composed of water, fat, lactose, important minerals, and protein mixtures, and are typically acidified to a pH between 3.0 and 4.6 [5]. The stability of proteins under acidic conditions is one of the most important factors in the development of acidified milk drinks.

The instability of milk proteins under acidic condition is generally believed to be caused by the structural instability and aggregation of casein micelles. The diameter of casein micelles ranges from 80 nm to 400 nm [6,7]. It is generally believed that the four subfractions of αs1-, αs2-, β-, and κ-caseins are bridged by calcium phosphate nanoclusters to form casein micelles [8]. The outer layer of casein micelles is composed of κ- and αs-caseins, whereas the inner layer is mainly composed of αs- and β-caseins [9,10]. The C terminal of κ-casein with a negative charge is hydrophilic and prevents casein from aggregating by electrostatic and steric repulsions [6]. However, when casein is acidified to a pH near the isoelectric point (at about 4.6), the κ-casein on the surface of casein micelles collapses. The lack of electrostatic and steric repulsions leads to the aggregation and sedimentation of casein [11,12].

For a long time, pectin, propylene glycol alginate, and carboxyl methyl cellulose (CMC) have been used to stabilize proteins in acidified milk drinks [13,14,15]. The isoelectric point of polysaccharide stabilizers (such as pectin) is relatively lower, generally at 3.6 [16]. When the pH is below the isoelectric point of casein and above the isoelectric point of polysaccharide, these negatively charged stabilizers are adsorbed on the specific parts of the casein with dominant positive charges to form complexes, which are essential to stabilize the acidified casein [17,18]. The ability of stabilizers to stabilize milk proteins depends not only on the ability to form complexes and the number of complexes but also on the electrostatic repulsion or steric hindrance between complexes. In addition, the viscosity of the mixed solutions and the presence of a weak gel effect also affect the stability of milk proteins [19,20].

Recently, soluble soybean polysaccharide (SSPS), a new polysaccharide extracted from soybean dregs, has attracted attention due to its ability to stabilize milk proteins under acidic conditions [3,16,21,22]. The structure of SSPS is similar to that of pectin. The main backbone of SSPS consists of homogalacturonan and rhamnogalacturonan, with the diglycosyl units, (1-4)-α-D-Gal A-(1-2)-α-L-Rha-(1-4), arranged repetitively [23]. The neutral sugar chains are composed of galactan and arabinan. In addition, a small percentage of hydrophobic proteins is connected to SSPS [24]. In contrast to pectin, SSPS has long side chains and a spherical structure, which may result in lower viscosity and better taste than high methoxyl pectin (HMP), guar gum, or CMC [25,26]. In addition, SSPS has better ability to stabilize casein than HMP at low pH (especially below 3.6) [25,27].

HMP has been shown to be adsorbed on casein micelles by electrostatic interaction during acidification, preventing the casein from aggregating through electrostatic and steric stabilization [15,20,28,29]. In contrast to HMP, it is suggested that SSPS prevents aggregation of casein mainly by the steric repulsion, and the electrostatic repulsion is auxiliary [3,28]. The electrostatic interaction between SSPS and casein plays a major role in the formation of complexes [30].

Previous studies have indicated that SSPS with high molecular weight (MW) and low degree of esterification (DE) has good ability to stabilize acidified milk drinks [31,32,33,34]. However, studies investigating the appropriate MW and DE of SSPS to stabilize acidified milk drinks showed different results. Li et al. (2009) reported that a high MW fraction of SSPS (5.5 × 10^5^ Da) had better ability to prevent aggregation and precipitation of proteins than a low MW fraction (5 × 10^3^ Da) [33]. Another study reported that a new SSPS with higher MW (2.85 × 10^6^ Da) could stabilize acidified milk drinks better than SSPS with a lower MW (5.5 × 10^5^ Da), and this was mainly due to the larger steric repulsion formed by complexes [14]. However, research indicated that the low MW SSPS (1.6 × 10^6^ Da) obtained through de-esterification still showed good stabilizing behavior, and a DE value between 25% and 17% was better for stabilizing due to the larger steric repulsion between complexes [31]. Xiong et al. (2015) found that the ability of SSPS to stabilize proteins increased as the DE value decreased from 83% to 49%, and SSPS with both lower DE and lower MW had better stabilizing ability [34]. These inconsistent results suggest that the mechanism of SSPS stabilizing acidified casein is still not clear, and the influence of structural factors such as MW and DE on the complexes needs to be explored further. Moreover, with a decrease in DE, the MW will also decrease, leading to the decreased ability to stabilize proteins. How to balance the combination of MW and DE becomes a challenge.

In this work, nine SSPSs extracted from soybean dregs with different MWs and DEs were used to explore the influence on stabilizing casein ability. The focus stability in this work was to protect casein from aggregation. The quantity of SSPS–casein complexes formed at different SSPS concentrations was quantified by size exclusion chromatography (SEC)-HPLC). The ability of SSPS with different MWs and DEs to stabilize casein was analyzed by the instability index, complex content, average particle size, zeta potential, and storage experiments.

## 2. Materials and Methods

### 2.1. Materials

The acid casein 741 was obtained from Fonterra Co-operative Group (Wellington, New Zealand), and contained 90% protein. CSSPS and soybean dregs were purchased from Pingdingshan Jinjing Biotechnology Co., Ltd. (Henan, China). Citric acid was obtained from Sino pharm Group Chemical Reagent Co., Ltd. (Shanghai, China).

### 2.2. Extraction and Characterization of SSPSs with Different MWs and DEs

#### 2.2.1. Extraction of SSPS

Nine SSPSs with different DEs and MWs were extracted from soybean dregs. First, soybean dregs were dispersed in distilled water at a ratio of 1/27, and SSPSs with different MWs were extracted under the acid treatment condition shown in Table 1. Then, the mixtures were centrifuged (4000× *g*, 20 °C) to separate the supernatant and precipitate. Ethanol was added to the supernatant at a volume ratio of 1/3, followed by centrifugation (4000× *g*, 20 °C) to obtain the SSPS sediments. The sediments were dried at 45 °C for 12 h to obtain SSPSs with different MWs. Degreasing was achieved under different alkali treatment conditions (Table 1). SSPSs with different MWs were dissolved in distilled water at a concentration of 10% (*w*/*v*), and the solutions were adjusted to specific pH values. Ethanol was added to solutions at a volume ratio of 1/3 to precipitate SSPS and the sediments were dried at 45 °C for 12 h. Nine SSPSs with different MWs and DEs were obtained sequentially—high MW and high DE (HH), high MW and medium DE (HM), high MW and low DE (HL), medium MW and high DE (MH), medium MW and medium DE (MM), medium MW and low DE (ML), low MW and high DE (LH), low MW and medium DE (LM), and low MW and low DE (LL).

#### 2.2.2. Measurement of MW and DE

MW was analyzed by analytical gel-filtration chromatography equipped with a Waters 2695 Separations Module and Waters 2410 Refractive Index Detector. Samples were analyzed using a TSK-gel 5000PWXL column (7.8 mm × 30 mm) in 50 mM sodium acetate buffer (pH 6.8).

DE was analyzed with titration. SSPS (500 mg) was dissolved in 100 mL of ethanol–HCl solution (12 M HCl:70% (*v*/*v*) ethanol = 1:19) and stirred for 10 min. Then, the SSPS solution was filtered and washed thoroughly six times using the ethanol–HCl mixture and 70% (*v*/*v*) ethanol, sequentially. The filter residue was washed using 20 mL of anhydrous ethanol and dried at 60 °C for 12 h. Then, 200 mg of dried sample was dissolved in 40 mL of distilled water with 0.8 mL of anhydrous ethanol with stirring. The SSPS solution was titrated by 0.1 M NaOH solution (V1) using phenolphthalein as an indicator. NaOH solution (8 mL, 0.5 M) was added to the SSPS solution with stirring for 30 min to demethylate. After stirring, 5 mL of HCl solution (0.5 M) was added to the SSPS solution to neutralize the NaOH solution. The SSPS solution was again titrated with 0.1 M NaOH solution (V2). The DE was calculated using Equation (1):DE=V_2_/(V_1_ + V_2_) × 100%(1)
where V1 and V2 are the initial and final volume of 0.1 M NaOH solution, respectively.

### 2.3. Preparation of SSPS–Casein Mixture

Casein was dissolved in distilled water at a concentration of 2.0% (*w*/*v*). The nine SSPSs and the commercial SSPS solutions were prepared using distilled water and stirred with a magnetic stirrer at 60 °C until dissolved. The casein and SSPS solutions were mixed at equal volume proportions. The mixtures were acidified to pH 4.0 by adding 40% citric acid solution at 25 °C, and the process of crossing the isoelectric point of casein was controlled within 10 s. The final concentrations of the mixtures were calculated to obtain mixtures with contents of 1% (*w*/*v*) casein, the ratios of SSPS to casein were 1/10, 2/10, 3/10, 4/10, 5/10, 6/10, 7/10, 8/10, 9/10, and 10/10. A mixture of 1.0% commercial SSPS and 1.0% casein at pH 7.0 was also prepared.

### 2.4. Stability Analysis

The stability of casein stabilized by nine SSPSs with different MWs and DEs was further analyzed by LUMiSizer (Dispersion Analyser LUMiSizer 651, LUM GmbH, Germany). LUMiSizer evaluated the stability of products through centrifugal acceleration technology based on the Lambert–Beer and Stokes’ laws [35].The instability index referred to changes of the transmission profile of the sample over time and space. The higher the instability index, the greater the change of the transmission profile of the sample after centrifugation, and the worse the stability of the sample.

Approximately 4 mL of sample was injected into a standard cuvette and centrifuged at 2150× *g* for 8400 s. For the measurement program, the time interval was 15 s and the temperature was 25 °C.

### 2.5. The Complex Content of SSPS–Casein Quantified by SEC-HPLC

The casein, commercial SSPS, and their mixture at pH 4.0 and 7.0 were quantified by SEC-HPLC. The HPLC was composed of a Waters 1525 binary HPLC pump, Waters 2487 dual λ absorbance detector, and Waters 2707 autosampler. The samples were analyzed by size exclusion columns (Shodex PROTEIN KW-804, 7 µm, 1500 Å, 8.0 mm inner diameter × 300 mm). The absorbance of ultraviolet radiation was detected at a wavelength of 280 nm. Phosphate buffers with sodium chloride were used as mobile phases. The mobile phase consisted of 50 mM phosphate and 0.3 M sodium chloride at pH 7.0, and 50 mM phosphate and 0.1 M sodium at pH 4.0. The mobile phase was filtered using a 0.45 μm cellulose membrane and degassed by ultrasonic cleaners for 30 min. The sampling volume was 10 µL, and the sampling time was 25 min at a flow rate of 1.0 mL/min. The UV wavelength was 220 nm. The temperature was set at 30 °C. All the prepared samples were stored at room temperature for 4 h before being injected.

The contents of the nine SSPS–casein complexes were measured as above.

### 2.6. Particle Size Distribution and Zeta Potential Characteristics

The particle size distribution and diameter of the mixtures were characterized using a Zetasizer Nano ZS, ZEN 3600 (Malvern Panalytical Ltd, Shanghai, China). The refractive index of the sample and dispersant were 1.590 and 1.330, respectively. Measurements were performed at 25 °C. All samples were diluted to 1/10 with 0.02 M citric acid buffer at the same pH value to avoid the effects of high concentrations of multiple scattering.

### 2.7. Storage Experiments

Samples obtained were kept at 4 °C for 30 days, and photos were recorded.

### 2.8. Statistics

All experiments were performed at least in triplicate, and drawings were performed using Origin 2017 (OriginLab Corporation, Northampton, Massachusetts, USA). Statistical significance was analyzed with Statistic software 9.0 (Analytical Software, Tallahassee, FL, USA). The data were reported as the means ± SD, a *p* value < 0.05 was statistically significant throughout the study.

## 3. Results

### 3.1. Properties of SSPS

The MW and DE of major fractions of nine SSPSs are shown in Table 1. HH had the largest MW (1.42 × 10^6^ Da), and LL had the smallest MW (4.7 × 10^5^ Da). The vigorous extraction conditions (such as high temperature, long heating time, and strong acid or alkali) accelerated the hydrolysis of SSPS. After de-esterification, the MW of HH (1.42 × 10^6^ Da) decreased from 1.42 × 10^6^ Da to 1.15 × 10^6^ Da (HM) and 1.06 × 10^6^ Da (HL), which demonstrated that both the DE and MW of SSPS decreased during de-esterification. Changes in MW and DE can affect the ability of SSPS to stabilize casein.

### 3.2. Stability Analysis

The stabilities of SSPS with different MWs and DEs were evaluated by LUMiSizer instability index. The instability index was calculated by the change in the transmission profile over time and space during centrifugation. For LH, LM, and LL SSPSs, the mixtures were not measured when the ratio was lower than 6/10 due to the sediment. The instability index of casein stabilized by the nine SSPSs is shown in Figure 1, and a lower instability index means better long-term physical stability [31].

Figure 1a shows the instability index of casein stabilized by SSPS with high MW and different DEs (HH, HM, and HL). For the HH, the instability index decreased from 0.97 to 0.59 with increasing ratio of SSPS. For SSPSs with de-esterification (HM and HL), the instability index decreased from 0.94 to 0.54 (minimum) when the SSPS/casein ratio increased to 6/10, and then the instability index increased slightly with further increase in the SSPS ratio. There was a small difference between HM and HL. At the same SSPS/casein ratio, HM and HL showed a lower instability index than HH. It was reported that SSPS with lower DE possessed more negative charges, which was beneficial for the stabilization behavior in acid milk drinks [31,34]. A similar result was found with a medium MW (Figure 1b) and MM or ML reached a minimum at a ratio of 5/10. For low MW (Figure 1c), when the SSPS/casein was below 5/10, the casein aggregated and precipitated. Differently from high and medium MW, LM showed the best stabilizing activity and LH the worst.

In terms of MW, comparing the SSPSs without de-esterification (HH, MH, and LH), the casein stabilized by SSPS with a medium MW (MH) of 8.8 × 10^5^ Da showed a lower instability index than HH (1.42 × 10^6^ Da) and LH (5.2 × 10^5^ Da). However, for medium or low DE, when the SSPS/casein ratio was below 5/10, there was little difference between SSPSs with high MW (HM and HL) and SSPSs with medium MW (MM and ML). When the ratio was >5/10, the instability index of SSPSs with high MW (MM and ML) was lower than SSPSs with medium MW (MM and ML). Meanwhile, the stabilizing ability of SSPSs with low MW at the same DE (LH, LM, and LL) to stabilize casein was worse than that with high or medium MW. Nobuhara et al. (2014) reported that SSPS with higher MW (2.85 × 10^6^ Da) had better ability to prevent casein aggregation than that with a lower MW (5.5 × 10^5^ Da) due to the strong steric repulsion of complexes [14]; which was different from the result in this study that HM had better stabilizing ability than HH, which implied the stabilizing activity of SSPS should consider both DE and MW.

The complexes formed by SSPS with high MW had larger steric repulsion; but high MW might make it difficult for SSPS to form complexes with casein. However, SSPS with low DE could not only provide larger electrostatic repulsion for complexes but also contribute to the electrostatic interaction between SSPS and casein. These results suggest that the ability of SSPS to stabilize proteins is not only related to the stability of the complexes themselves but might also be associated with the ability to form complexes.

### 3.3. Complex Content Quantified by SEC-HPLC

#### 3.3.1. Complex Content of CSSPS–Casein at Different pH Values

The complexes formed by SSPS and casein under acidic condition were quantified to investigate the influence of MW and DE on the ability of SSPS to combine casein. Zhao, Li, Carvajal, and Harris (2009) reported that the polysaccharide–protein complexes driven by electrostatic interactions could be detected using SEC-HPLC [36]. A Waters 2487 dual λ absorbance detector was used to investigate the interaction between SSPS and casein at different pH values due to the hydrophobic protein in SSPS [24].

Figure 2 shows the chromatograms of SSPS, casein, and their mixtures at pH 4.0 and 7.0. As can be seen, the casein showed a major peak at the elution time of 11.7 min and commercial SSPS had a major peak at the elution time of 6.1 min. At the same concentration, the casein had much higher UV absorption than commercial SSPS. At pH 7.0, there was a slight difference in the chromatogram between the mixture and casein due to the tiny UV absorption for the commercial SSPS. At pH 4.0, a large new peak appeared at the elution time of 5.8 min, compared with the curve of SSPS and casein. The UV absorption value (area) of the complex (2.9 × 10^6^) was much larger than that of SSPS (7.4 × 10^4^) and casein (1.1 × 10^5^). This confirmed that complexes were formed between positively charged SSPS and negatively charged casein at pH 4.0, and only the content of complexes was detected, as reflected by the UV absorption value (area) at the elution time of 5.8 min.

This result indicated that the casein and SSPS formed complexes mainly through electrostatic interaction. At pH 7.0, there was little interaction between SSPS and casein, which may be because both SSPS and casein had negative charge. When acidified to pH 4.0, which is below the isoelectric point, the casein surface became positively charged [11]. Meanwhile, the SSPS was still positively charged due to the large amount of carboxyl groups on the main chain. The SSPS was adsorbed on the casein surface to form a complex through electrostatic interactions. The results correspond to a previous report that SSPS did not interact with caseins at neutral pH [25,37].

#### 3.3.2. Complex Content of Nine SSPS–Casein Complexes at Different Concentrations Quantified by SEC-HPLC

The UV absorption value (area) at the elution time of 5.8 min of SEC chromatograms reflected the SSPS–casein complex content. The higher UV absorption value (area) reflected more SSPS–casein complexes formed, which was beneficial to stabilize casein under acid conditions.

The content of complexes (Figure 3) showed similar results to those in Section 3.2. At the same SSPS MW, the quantity of complexes formed by SSPS with low DE and casein was the highest, which indicated the best stability. However, SSPS without de-esterification formed fewer complexes with casein. It has been reported that SSPS with low DE had good stabilizing behavior in acid milk drinks [31,34]. It was considered that the complexes formed by casein and SSPS with low DE had more negative charges, and the stronger electrostatic repulsion prevented casein aggregation better [14,31]. These results indicated that the increasing electric charge of SSPS during de-esterification contributed to the electrostatic interaction between casein and SSPS, and more complexes provided SSPS with better ability to stabilize casein.

At the same DE, for SSPS without de-esterification, SSPS with medium MW (MH) was easily adsorbed on the casein surface to form more complexes than SSPS with high or low MW (HH and LH). Compared with SSPS with high MW (HM and HL), SSPS with medium MW (MM and ML) formed more complexes with casein and showed similar instability indexes as before at a ratio of 5/10. Moreover, HM and HL had better stability than MM and ML as the ratio of SSPS increased, which was consistent with the results in Section 3.2. This suggested that SSPS with high MW could generate a larger steric repulsion to stabilize casein, even though fewer complexes were formed. These results were similar to a previous study that showed that SSPS-HMW (high-molecular-mass complex of SSPS cross-linked via phosphate) had better stabilizing ability due to the stronger steric repulsion of the thicker layer formed on the protein surface [14,38]. LM and LL showed a low stabilizing ability when a large number of SSPS–casein complexes were formed, which was due to the weak repulsion between complexes.

For HM and HL, the ratio of SSPS/casein showed some specific results. The SSPS–casein complex content reached a maximum (Figure 3a) and casein showed the best stability (Figure 2a) at a ratio of 6/10. However, with increased ratio of SSPS, the quantity of the complexes remained constant and the stability decreased slightly. This indicated that the casein was completely covered by SSPS at the saturation ratio 6/10. When the ratio was greater than the saturation ratio, the excess SSPS might be adsorbed in multilayers on the surface of the complexes and would not contribute to the formation of complexes, leading to the decrease in stability. A similar result was found for MM and ML when the saturation ratio was 5/10.

The previous study reported that MW or DE was essential to the stabilizing ability of SSPS due to their effects on steric or electrostatic repulsion between complexes [14,31,33,38]. Combining with the result of Section 3.2, the ability of SSPS to be absorbed on the surface of casein to form complex was also important. The ability of SSPS to form complexes with casein can be improved by de-esterification, and the instability can also decrease. The steric repulsion provided by the side chain of SSPS was important for the stability of casein.

### 3.4. Particle Size Distribution

As shown in Table 2, the Z-average size of SSPS–casein mixtures was obtained by dynamic light scattering technique based on light intensity. The polymer dispersity index (PDI) value of SSPS–casein mixtures (Table 3) was used to describe the uniformity of particle size distribution, and a smaller value indicated a more homogeneous polymer. According to Stokes’ law, a larger size of particles leads to precipitation of protein.

At the same MW, SSPS after de-esterification showed a smaller Z-average size and a lower PDI due to the formation of more complexes, which corresponded to the instability index (Section 3.2).

In terms of the effect of MW on Z-average size, for SSPS without de-esterification, the casein stabilized by MH was smaller than with HH and LH because more complexes were formed. At the medium and low DE, there was a small difference between SSPS with high MW and medium MW. However, the casein stabilized by LM and LL showed small Z-average size and low PDI until the SSPS/casein ratio was greater than 6/10, due to the weak steric repulsion between complexes.

For HL, the Z-average size and PDI of casein decreased to a minimum at the saturation ratio. Meanwhile, LL formed most complexes and showed the lowest instability index at this ratio (Section 3.2 and Section 3.3.2). These results indicated that SSPS–casein complexes can prevent the aggregation of casein, which resulted in a more uniform particle size distribution and smaller mean particle diameter. In addition, the smaller Z-average size indicated that the complexes contained fewer casein micelles [31]. When the SSPS/casein ratio was over the saturation ratio, the extra SSPS did not form complex any more, while LUMiSizer instability index and Z-average size slight increased, which implied extra SSPS might be absorbed on the surface of complex in multilayer.

### 3.5. Zeta Potential Properties

The zeta potential of casein, commercial SSPS, and their mixtures at different pH values is shown in Table 4. The zeta potentials of commercial SSPS and casein were around −20 mV and −38 mV at pH 7.0, respectively. There were two peaks of zeta potential in the mixture, corresponding to casein and commercial SSPS. However, there was only one zeta potential peak for the mixture at pH 4.0 compared with that at neutral pH. These results indicated that there was almost no interaction between casein and SSPS at the neutral conditions, whereas SSPS was adsorbed on the casein surface to form complexes under acid conditions. This was consistent with the results in Section 3.3.1.

The zeta potential of casein with the nine SSPSs (Table 5) showed similar results. With increasing ratio of SSPS, casein with a positive charge was covered by negatively charged SSPS, and the zeta potential changed from positive to negative. The absolute value of zeta potential for all samples remained a low level and LL had the lowest zeta potential. However, LL showed bad stabilizing ability, which was different from a previous report that SSPS with low DE stabilized casein better due to the larger electrostatic repulsion between complexes [31,34]. This difference may be related to the weak steric repulsion between complexes formed by LL and casein. The absolute value of zeta potential of these complexes was much lower than the −16 mV reported by Cai et al. (2020), while these complexes still had a high stability. These results indicated that the de-esterification enhanced the stabilizing ability of SSPS mainly by contributing to the formation of complexes; SSPS prevented the aggregation of casein mainly through strong steric repulsion, and the electrostatic repulsion between complexes was weak.

### 3.6. Storage Stability

The stability of casein stabilized by SSPS with different MWs and DEs after storing at 4 °C for 30 days is shown in Figure A1. At the same MW, the light transmittance of SSPS with low and medium DE was significantly higher than that of SSPS without de-esterification, which was consistent with the results for the content of complex and the instability index. However, at a ratio of 10/10, HH and HL had the same instability index, but the transmittance of casein stabilized by HH was significantly worse than that with HL.

For SSPS without de-esterification, the casein stabilized by MH was more homogeneous, whereas precipitates could be easily seen for SSPS stabilized by LH. For low and medium DE, when the ratio was 2/10 and 3/10, the SSPS with medium MW had better stability. When the ratio exceeded 4/10, there was no significant difference between the high and low MW. For LM and LL, the results were similar to those for the instability index. When the ratio of SSPS/casein was below 5/10, casein precipitated completely. When the ratio was greater than 6/10, the transmittances of the solutions were lower although the solutions were homogeneous.

For HL, there was a large amount of precipitation when the ratio of SSPS/casein was below 2/10. With increasing ratio of SSPS, the amount of precipitation decreased and the light transmittance increased, which might be related to the decreasing particle size of the complexes. However, in contrast to the results for the instability index, excess SSPS could not reduce the stability although the particle size increased. The same results were found for HM, MM, and ML.

Combined with the instability index and the complex content, the long-term stability of casein stabilized by SSPS needed to be measured with the content of complex and the stability of the complexes. For the complex with good stability, only a small amount of complex could provide excellent stabilizing ability. For the complex with weak stability, a large amount of complex was needed. The more complexes the solution had, the more stabilized the solution was and the higher the transmittance.

## 4. Conclusions

The ability of SSPS to stabilize casein was related to the content of SSPS–casein complexes and the repulsion between complexes. At pH 4.0, SSPS was adsorbed on the surface of casein to form complexes before casein aggregated, which could be detected by SEC-HPLC. With regard to the ability of SSPS at the same MW to combine casein, the SSPS with medium or low DE could form more complexes than SSPS with high DE due to more negative charge on the main chain. At the same DE, the SSPS with medium MW was adsorbed on the casein surface more easily and formed more complexes with casein. The stability of the complexes themselves was also an important factor when a large number of complexes was formed. This was mainly due to the strong steric repulsion of the complexes and the weak electrostatic repulsion between complexes. The increasing number of complexes with good stability could effectively reduce the average particle size and improve the stability. The excess SSPS could not increase the complex content, and this might be related to further multilayer adsorption of SSPS on the casein surface. This could increase the particle size of complexes but have no negative effect on the long-term stability. For the complex with good stability, only a small amount of complex provided excellent stabilizing ability. For the complex with weak stability, a large amount of complex was needed. The more complexes the solution had, the more stabilized the solution was and the higher the transmittance.

These results showed that, considering the complex content and the stability of complexes, the SSPS with high MW and low DE (HL) showed the best ability to stabilize casein under acid conditions. These results contribute to a better understanding of the interactions between SSPS and casein and the stabilizing of acidified milk protein.

## Figures and Tables

**Figure 1 foods-10-00686-f001:**
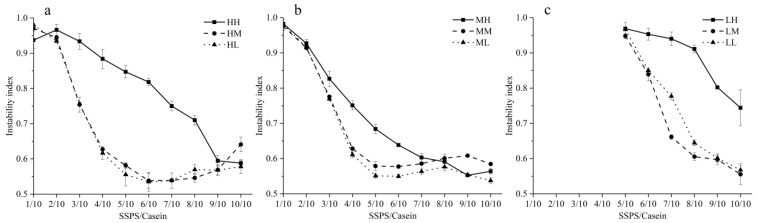
The instability index of casein stabilized by SSPS with high molecular weight (MW) (**a**), medium MW (**b**), and low MW (**c**).

**Figure 2 foods-10-00686-f002:**
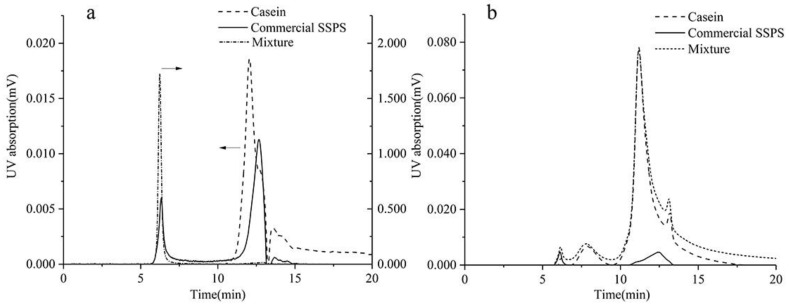
The UV absorption of commercial SSPS, casein, and their mixtures at pH 4.0 (**a**) and 7.0 (**b**) of size exclusion chromatography (SEC) chromatogram.

**Figure 3 foods-10-00686-f003:**
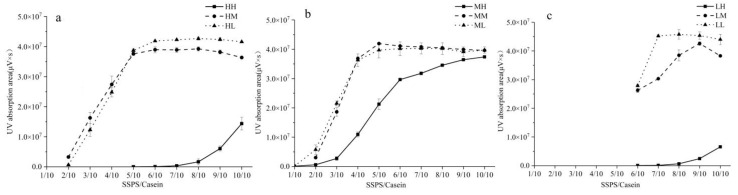
The UV absorption of complexes formed by casein and SSPS with high MW (**a**), medium MW (**b**), and low MW (**c**), which was quantified by SEC-HPLC.

**Table 1 foods-10-00686-t001:** The extraction conditions and properties of the nine soluble soybean polysaccharides (SSPSs).

SSPS	Extraction Conditions			Degreasing Conditions			MW (×104 Da)	DE (%)
	pH	Temperature (°C)	Time (h)	pH	Temperature (°C)	Time (h)		
HH	3.4	100	2	–	–	–	142	62
HM				12	90	2	115	37
HL				12.5	90	2.5	106	20
MH	4	120	2	–	–	–	88	64
MM				12	90	2	81	40
ML				12.5	90	2.5	72	30
LH	3.2	120	3	–	–	–	52	54
LM				12	90	2	48	35
LL				12.5	90	2.5	47	26

**Table 2 foods-10-00686-t002:** The Z-average particle size (nm) of the SSPS–casein mixtures.

SSPS	Ratio of SSPS to Casein (*w*/*w*)								
	2/10	3/10	4/10	5/10	6/10	7/10	8/10	9/10	10/10
HH	459.0 ± 4.5 ^a^	401.6 ± 6.2 ^b^	401.2 ± 1.8 ^b^	324.6 ± 0.9 ^c^	312.9 ± 2.2 ^c^	294.9 ± 5.1 ^c^	293.5 ± 3.7 ^c^	322.2 ± 13.6 ^c^	302.1 ± 26.2 ^c^
HM	296.2 ± 5.9 ^a^	172.4 ± 0.2 ^b^	140.0 ± 1.7 ^c^	132.8 ± 1.5 ^c,d^	132.5 ± 0.5 ^d^	132.0 ± 0.6 ^d^	124.4 ± 1.5 ^e^	125.7 ± 1.5 ^d,e^	132.4 ± 0.8 ^d^
HL	329.9 ± 12.0 ^a^	185.0 ± 1.9 ^b^	148.0 ± 0.8 ^c^	138.0 ± 0.6 ^c,d^	128.1 ± 0.8 ^d^	125.2 ± 1.2 ^d^	131.1 ± 1.2 ^d^	141.5 ± 1.2 ^c^	139.1 ± 1.0 ^c,d^
MH	258.3 ± 0.1 ^a^	205.7 ± 0.4 ^b^	202.2 ± 2.4 ^b^	181.7 ± 2.3 ^c,d^	180.7 ± 0.3 ^d,e^	172.6 ± 1.8 ^e^	187.7 ± 1.0 ^c,d^	190.0 ± 4.5 ^c^	202.8 ± 5.4 ^b^
MM	238.3 ± 3.8 ^a^	171.8 ± 1.5 ^b^	146.3 ± 1.0 ^c^	134.8 ± 0.2 ^d,e^	131.6 ± 1.1 ^e^	132.6 ± 0.7 ^d e^	133.5 ± 0.3 ^d,e^	137.2 ± 1.2 ^d^	148.0 ± 1.4 ^c^
ML	239.8 ± 4.5 ^a^	161.4 ± 2.8 ^b^	145.2 ± 1.4 ^c^	134.9 ± 0.8 ^d^	131.5 ± 0.1 ^d^	131.0 ± 1.6 ^d^	135.0 ± 1.2 ^d^	134.4 ± 3.2 ^d^	145.5 ± 1.0 ^c^
LH	–	–	–	–	525.5 ± 7.8 ^a^	491.5 ± 16.8 ^b^	319.5 ± 1.1 ^c^	259.7 ± 4.1 ^d^	216.8 ± 2.2 ^e^
LM	–	–	–	–	143.7 ± 0.5 ^d^	156.4 ± 0.3 ^b^	152.2 ± 0.1 ^c^	168.9 ± 1.0 ^a^	153.8 ± 0.2 ^c^
LL	–	–	–	–	234.7 ± 0.8 ^a^	188.8 ± 2.8 ^b^	168.0 ± 1.3 ^c^	146.4 ± 3.6 ^d^	148.6 ± 1.0 ^d^

Values are expressed as mean ± SD. Different superscripts (a–e) in the same row indicate significant differences (*p* < 0.05).

**Table 3 foods-10-00686-t003:** The PDI value of the SSPS–casein mixtures.

SSPS	Ratio of SSPS to Casein (*w/w*)								
	2/10	3/10	4/10	5/10	6/10	7/10	8/10	9/10	10/10
HH	0.71 ± 0.04 ^a^	0.47 ± 0.01 ^b^	0.45 ± 0.01 ^b^	0.40 ± 0.04 ^b^	0.38 ± 0.01 ^b^	0.38 ± 0.01 ^b^	0.40 ± 0.01 ^b^	0.42 ± 0.03 ^b^	0.39 ± 0.12 ^b^
HM	0.40 ± 0.02 ^a^	0.16 ± 0.01 ^c^	0.15 ± 0.02 ^c^	0.15 ± 0.01 ^c^	0.19 ± 0.01 ^bc^	0.22 ± 0.02 ^b^	0.18 ± 0.01 ^bc^	0.19 ± 0.01 ^bc^	0.21 ± 0.01 ^b^
HL	0.47 ± 0.05 ^a^	0.19 ± 0.01 ^bcd^	0.16 ± 0.01 ^d^	0.17 ± 0.01 ^cd^	0.20 ± 0.01 ^bcd^	0.20 ± 0.01 ^bcd^	0.21 ± 0.01 ^bcd^	0.24 ± 0.01 ^b^	0.23 ± 0.01 ^bc^
MH	0.23 ± 0.02 ^a^	0.19 ± 0.01 ^ab^	0.17 ± 0.01 ^abc^	0.17 ± 0.01 ^abc^	0.16 ± 0.01 ^abc^	0.20 ± 0.01 ^abc^	0.20 ± 0.01 ^bc^	0.20 ± 0.02 ^bc^	0.21 ± 0.02 ^c^
MM	0.27 ± 0.03 ^a^	0.17 ± 0.01 ^cd^	0.16 ± 0.01 ^d^	0.16 ± 0.02 ^d^	0.19 ± 0.02 ^bcd^	0.21 ± 0.01 ^bcd^	0.23 ± 0.01 ^ab^	0.23 ± 0.01 ^ab^	0.22 ± 0.01 ^abc^
ML	0.34 ± 0.03 ^a^	0.16 ± 0.02 ^de^	0.14 ± 0.02 ^e^	0.17 ± 0.02 ^cde^	0.19 ± 0.01 ^cde^	0.21 ± 0.01 ^bcd^	0.23 ± 0.01 ^bc^	0.21 ± 0.02 ^bcd^	0.26 ± 0.02 ^b^
LH	-	-	-	-	0.85 ± 0.02 ^a^	0.67 ± 0.08 ^b^	0.37± 0.01 ^c^	0.25 ± 0.02 ^cd^	0.21 ± 0.01 ^d^
LM	-	-	-	-	0.13 ± 0.01 ^a^	0.13 ± 0.02 ^a^	0.14 ± 0.01 ^a^	0.12 ± 0.01 ^a^	0.13 ± 0.03 ^a^
LL	-	-	-	-	0.23 ± 0.01 ^a^	0.16 ± 0.03 ^ab^	0.11 ± 0.03 ^b^	0.13 ± 0.03 ^b^	0.14 ± 0.02 ^ab^

Values are expressed as mean ± SD. Different superscripts (a–e) in the same row indicate significant differences (*p* < 0.05).

**Table 4 foods-10-00686-t004:** The zeta potential of commercial SSPS, casein, and their mixtures at pH 4.0 and 7.0.

Samples	pH	
	4.0	7.0
Casein	6.76 ± 0.18	−37.28 ± 1.04
Commercial SSPS	−7.37 ± 0.31	−21.65 ± 1.65
Mixture	−6.45 ± 0.34	−38.25 ± 0.69
		−23.20 ± 0.64

**Table 5 foods-10-00686-t005:** The zeta potential of the SSPS–casein mixtures.

SSPS	Ratio of SSPS to Casein (*w*/*w*)								
	2/10	3/10	4/10	5/10	6/10	7/10	8/10	9/10	10/10
HH	0.80 ± 0.05 ^a^	0.29 ± 0.22 ^b^	0.24 ± 0.05 ^b^	0.00 ± 0.4 ^b^	−0.16 ± 0.06 ^bc^	−0.52 ± 0.04 ^c^	−1.27 ± 0.02 ^d^	−1.48 ± 0.06 ^d^	−2.27 ± 0.03 ^e^
HM	0.60 ± 0.13 ^a^	−1.03 ± 0.22 ^c^	−0.46 ± 0.04 ^b^	−0.98 ± 0.00 ^c^	−1.55 ± 0.19 ^d^	−2.20 ± 0.02 ^e^	−2.00 ± 0.08 ^e^	−2.24 ± 0.06 ^e^	−2.24 ± 0.23 ^e^
HL	0.78 ± 0.10 ^a^	0.23 ± 0.06 ^b^	−0.36 ± 0.01 ^c^	−0.99 ± 0.06 ^d^	−1.11 ± 0.05 ^d^	−1.06 ± 0.11 ^d^	−1.13 ± 0.33 ^d^	−1.75 ± 0.11 ^e^	−1.28 ± 0.16 ^d^
MH	0.89 ± 0.08 ^a^	0.54 ± 0.07 ^ab^	0.39 ± 0.17 ^b^	−0.23 ± 0.20 ^c^	−0.97 ± 0.03 ^d^	−1.49 ± 0.08 ^e^	−2.01 ± 0.03 ^f^	−2.48 ± 0.14 ^g^	−3.13 ± 0.07 ^h^
MM	0.58 ± 0.20 ^a^	−0.30 ± 0.13 ^b^	−0.88 ± 0.01 ^c^	−1.71 ± 0.03 ^d^	−2.02 ± 0.05 ^de^	−2.45 ± 0.15 ^f^	−2.41 ± 0.07 ^f^	−2.93 ± 0.09 ^g^	−2.27 ± 0.00 ^ef^
ML	0.53 ± 0.10 ^a^	−0.19 ± 0.06 ^b^	−0.72 ± 0.10 ^c^	−1.71 ± 0.05 ^d^	−1.73 ± 0.10 ^d^	−1.94 ± 0.05 ^de^	−2.13 ± 0.06 ^ef^	−2.37 ± 0.29 ^f^	−2.95 ± 0.03 ^g^
LH	–	–	–	–	−0.85 ± 0.04 ^a^	−0.84 ± 0.20 ^a^	−1.70 ± 0.03 ^b^	−2.44 ± 0.04 ^c^	−3.24 ± 0.28 ^d^
LM	–	–	–	–	−3.29 ± 0.09 ^a^	−4.12 ± 0.33 ^b^	−4.40 ± 0.26 ^bc^	−4.95 ± 0.27 ^c^	−4.76 ± 0.02 ^bc^
LL	–	–	–	–	−5.00 ± 0.18 ^a^	−4.67 ± 0.54 ^a^	−5.50 ± 0.28 ^ab^	−5.41 ± 0.03 ^a^	−6.57 ± 0.28 ^b^

Values are expressed as mean ± SD. Different superscripts (a–h) in the same row indicate significant differences (*p* < 0.05).

## Data Availability

Data is contained within the article.

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
