# Peer review of "Effects of Molecular Weight and Degree of Esterification of Soluble Soybean Polysaccharide on the Stability of Casein under Acidic Conditions"

_foods, 2021, doi:10.3390/foods10030686_

Round 1
Reviewer 1 Report
Overall an interesting piece of work by the authors where they combine a good characterization of SSPS with a good characterization of the systems in which they use it. Often in papers I see one or the other, but in this case it’s nice to see both. There are, however, some points that need attention:
-line 39: ‘stretches into the center’ does not make sense
-line 43: reverse the order. Sedimentation results from aggregation, not the other way around
-line 44-53: in discussing these aspects, keep in mind that it is about specific parts of the protein with dominant positive or negative charges, not about the overall protein
-line 96: ‘Casein’ needs to be specified. Fonterra sells many different types of casein so just mentioning ‘casein’ is not enough
-line 140: need to include the temperature at which acidification was done
-Line 141: how quickly? And of course, you’re still quite close to the IEP
-Line 149: express in g rather than rpm
-Line 150: data analysis should be included and explained so that the reader knows what the instability index used later actually means
-Line 169: the ‘citrate/phosphate buffer’ should be explained better
-Section 3.3.2.: the whole interpretation of UV absorption should be explained better and the figure legends should make clear that these are SEC chromatograms and not simple UV absorption measurements
-Line 205: change ‘an’ to ‘a’
-Line 312-314: not clear why you would expect multilayer absorption. Please explain
-Line 315: values reported are not distributions
-Line 370: this is more a conclusion than a discussion. In general the paper lacks discussion in my opinion and has only very limited referencing to previous work. This clearly needs to be improved!!!!
Reviewer 2 Report
General:
The subject matter of the work is within the scope of the Foods paper, but some things are not fully developed.
Specific comments:
- In the discussion chapter, there is no reference to other studies in this field, which should be supplemented.
- The charts (numbers 1, 2, 3) are unreadable and need improvement.
Reviewer 3 Report
This paper deals with the aggregation protection by soybean polysaccharide derivatives to casein. The paper contains interesting results.
— The word “stability” contains several kinds of meanings such as heat stability, pH stability, and molecular-level conformation stability. Among the meanings, the authors should describe the focus stability is the maintain of protection from aggregation. It should be clearly shown in Introduction.
— There are some related researches for this purpose. Especially, reference 31) Cai et al. is a quite similar contents compared with this paper. So, the authors should describe the difference points of this research in discussion section. Especially, the authors should emphasize the advantage points or novel knowledge points of this research compared with other researches.
